# Learning from Demonstration with Weakly Supervised Disentanglement

**Yordan Hristov**
School of Informatics
University of Edinburgh
`yordan.hristov@ed.ac.uk`

**Subramanian Ramamoorthy**
School of Informatics
University of Edinburgh
`s.ramamoorthy@ed.ac.uk`

## Abstract

Robotic manipulation tasks, such as wiping with a soft sponge, require control from multiple rich sensory modalities. Human-robot interaction, aimed at teaching robots, is difficult in this setting as there is potential for mismatch between human and machine comprehension of the rich data streams. We treat the task of interpretable learning from demonstration as an optimisation problem over a probabilistic generative model. To account for the high-dimensionality of the data, a high-capacity neural network is chosen to represent the model. The latent variables in this model are explicitly aligned with high-level notions and concepts that are manifested in a set of demonstrations. We show that such alignment is best achieved through the use of labels from the end user, in an appropriately restricted vocabulary, in contrast to the conventional approach of the designer picking a prior over the latent variables. Our approach is evaluated in the context of two table-top robot manipulation tasks performed by a PR2 robot – that of dabbing liquids with a sponge (forcefully pressing a sponge and moving it along a surface) and pouring between different containers. The robot provides visual information, arm joint positions and arm joint efforts. We have made videos of the tasks and data available - see supplementary materials at: https://sites.google.com/view/weak-label-lfd.

## 1 Introduction

Learning from Demonstration (LfD) (Argall et al., 2009) is a commonly used paradigm where a potentially imperfect demonstrator desires to teach a robot how to perform a particular task in its environment. Most often this is achieved through a combination of kinaesthetic teaching and supervised learning—imitation learning (Ross et al., 2011). However, such approaches do not allow for elaborations and corrections from the demonstrator to be seamlessly incorporated. As a result, new demonstrations are required when either the demonstrator changes the task specification or the agent changes its context—typical scenarios in the context of interactive task learning (Laird et al., 2017). Such problems mainly arise because the demonstrator and the agent reason about the world by using *notions* and mechanisms at *different levels of abstraction*. An LfD setup, which can accommodate abstract user specifications, requires establishing a mapping from the high-level notions humans use—e.g. spatial concepts, different ways of applying force—to the low-level perceptive and control signals robot agents utilise—e.g. joint angles, efforts and camera images. With this in place, any constraints or elaborations from the human operator must be mapped to behaviour on the agent's side that is consistent with the semantics of the operator's desires.

Concretely, we need to be able to ground (Vogt, 2002; Harnad, 1990) the specifications and symbols used by the operator in the actions and observations of the agent. Often the actions and the observations of a robot agent can be high-dimensional—high DoF kinematic chains, high image resolution, etc.—making the symbol grounding problem non-trivial. However, the concepts we need to be able to ground lie on a much-lower-dimensional manifold, embedded in the high-dimensional data space (Fefferman et al., 2016). For example, the concept of pressing softly against a surface manifests itself in a data stream associated with the 7 DoF real-valued space of joint efforts, spread across multiple time steps. However, the essence of what differentiates one type of soft press from another nearby concept can be summarised conceptually using a lower-dimensional abstract space. The focus of this work is finding a nonlinear mapping (represented as a high-capacity neural model) between such a

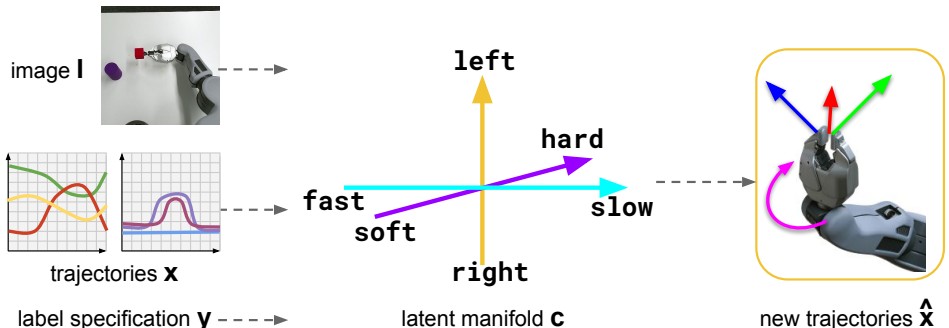

Figure 1: User demos through teleoperation and a variety of modalities **(left)** are used to fit a common low-level disentangled manifold **(middle)** which contributes in an interpretable way to the generative process for new robot behaviour **(right)**.

low-dimensional manifold and the high-dimensional ambient space of cross-modal data. Moreover, we show that, apart from finding such a mapping, we can also shape and refine the low-dimensional manifold by imposing specific biases and structures on the neural model's architecture and training regime.

In this paper, we propose a framework that allows human operators to teach a PR2 robot about different spatial, temporal and force-related aspects of a robotic manipulation task, using tabletop dabbing and pouring as our main examples. They serve as concrete representative tasks that incorporate key issues specific to robotics (e.g. continuous actions, conditional switching dynamics, forceful interactions, and discrete categorizations of these). Numerous other applications require the very same capability. Our main contributions are:

- A learning method which incorporates information from multiple high-dimensional modalities—vision, joint angles, joint efforts—to instill a disentangled low-dimensional manifold (Locatello et al., 2019). By using weak expert labels during the optimisation process, the manifold eventually aligns with the human demonstrators' '*common sense*' notions in a natural and controlled way without the need for separate post-hoc interpretation.

- We release a dataset of subjective concepts grounded in multi-modal demonstrations. Using this, we evaluate whether discrete latent variables or continuous latent variables, both shaped by the discrete user labels, better capture the demonstrated continuous notions.

## 2   IMAGE-CONDITIONED TRAJECTORY & LABEL MODEL

In the task we use to demonstrate our ideas, which is representative of a broad class of robotic manipulation tasks, we want to control **where** and **how** a robot performs an action through the use of a user specification defined by a set of coarse labels—e.g. *"press softly and slowly behind the cube in the image"*. In this context, let, $\mathbf{x}$ denote a $K \times T$ dimensional trajectory for $K$ robot joints and a fixed time length $T$, $\mathbf{y}$ denote a set of discrete labels semantically grouped in $N$ label groups $\mathcal{G} = \{g_1, \ldots, g_N\}$ (equivalent to multi-label classification problem) and $\mathbf{i}$ denote an RGB image[1]. The labels $\mathbf{y}$ describe qualitative properties of $\mathbf{x}$ and $\mathbf{x}$ with respect to $\mathbf{i}$—e.g. `left` dab, `right` dab, `hard` dab, `soft` dab, etc. We aim to model the distribution of demonstrated robot-arm trajectories $\mathbf{x}$ and corresponding user labels $\mathbf{y}$, conditioned on a visual environment context $\mathbf{i}$. This problem is equivalent to that of structured output representation (Walker et al., 2016; Sohn et al., 2015; Bhattacharyya et al., 2018)—finding a one-to-many mapping from $\mathbf{i}$ to $\{\mathbf{x}, \mathbf{y}\}$ (one image can be part of the generation of many robot trajectories and labels). For this we use a conditional generative model, whose latent variables $\mathbf{c} = \{\mathbf{c}_s, \mathbf{c}_e, \mathbf{c}_u\}$ can accommodate the aforementioned mapping—see Figure 2. The *meaning* behind the different types of latent variables—$\mathbf{c}_s, \mathbf{c}_e$ and $\mathbf{c}_u$—is elaborated in section 3.

---

[1]What $\mathbf{i}$ actually represents is a lower-dimensional version of the original RGB image $\mathbf{I}$. The parameters of the image encoder are jointly optimised with the parameters of the recognition and decoder networks

We want the variability in the trajectory and label data to be concisely captured by $\mathbf{c}$. We choose by design the dimensionality of the latent variables to be much lower than the dimensionality of the data. Therefore, $\mathbf{c}$ is forced to represent abstract properties of the robot behaviour which still carry enough information for the behaviour to be usefully generated—absolute notions like speed, effort, length, and relative spatial notions which are grounded with respect to the visual context. Another way to think of $\mathbf{c}$ is as a continuous version of $\mathbf{y}$ which can therefore incorporate nuances of the same label. In the existing literature discrete labels are usually represented as discrete latent variables—e.g. digit classes (Kingma et al., 2014).

However, the discrete labels humans use are a rather crude approximation to the underlying continuous concepts—e.g. we may have the notion for a `soft` and a `softer` dab even though both would be alternatively labelled as `soft`. For this reason, we use continuous latent variables, shaped by discrete labels, to represent these subjective concepts.

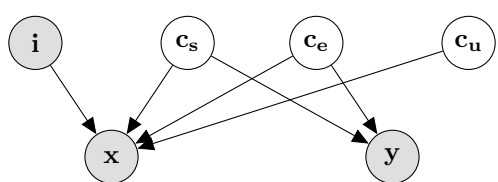

Figure 2: The lower-dimensional encoding $\mathbf{i}$ of the environment context $\mathbf{I}$ is observed. Conditioned on $\mathbf{i}$ and the latent variables $\mathbf{c}$, sampled from the prior over $\mathbf{c}$, we get a distribution over possible robot trajectories $\mathbf{x}$ and user labels $\mathbf{y}$.

The joint distribution over $\mathbf{x}$ and $\mathbf{y}$, conditioned on $\mathbf{i}$, is modelled according to Eq. 1. A prior of choice—isotropic Gaussian, Uniform, Categorical—is placed over the latent variables $\mathbf{c}$ which are independent of the image.

$$p(\mathbf{x}, \mathbf{y}|\mathbf{i}) = \int p(\mathbf{x}, \mathbf{y}|\mathbf{i}, \mathbf{c})p(\mathbf{c})d\mathbf{c} = \int p(\mathbf{x}|\mathbf{c}, \mathbf{i})p(\mathbf{y}|\mathbf{c})p(\mathbf{c})d\mathbf{c} \tag{1}$$

We choose to represent the distribution over $\mathbf{x}$ and $\mathbf{y}$ as Gaussian and Multinomial distributions respectively. The parameters of these distributions are defined as nonlinear functions (of the image context and latent variables) which are represented as the weights $\theta$ of a neural network $p_\theta$—$\boldsymbol{\mu}_\theta(\cdot)$ is a vector of means, $\boldsymbol{\sigma}_\theta(\cdot)$ is a vector of log variances and $\boldsymbol{\pi}_\theta(\cdot)$ is a vector of probabilities.

$$p_\theta(\mathbf{x}|\mathbf{c}, \mathbf{i}) = \mathcal{N}(\mathbf{x}|\boldsymbol{\mu}_\theta(\mathbf{c}, \mathbf{i}), \boldsymbol{\sigma}_\theta(\mathbf{c}, \mathbf{i})) \quad (2) \qquad p_\theta(\mathbf{y}|\mathbf{c}) = \mathrm{Cat}(\mathbf{y}|\boldsymbol{\pi}_\theta(\mathbf{c})) \tag{3}$$

The parameters of $p_\theta$ are optimised by maximising the variational lower bound (VLB) of the data distribution—see Eq. 4. We additionally optimise a recognition network $q_\phi$, parametrised by $\phi$, which acts as an approximation to the posterior of $\mathbf{c}$, also modelled as a Gaussian. The recognition network $q_\phi$ is used to find good candidates of $\mathbf{c}$, for given $\mathbf{x}$ and $\mathbf{i}$, thus making the optimisation of $\theta$, tractable (Kingma & Welling, 2013). The posterior is conditioned on the environment context as well, since $\mathbf{i}$ and $\mathbf{c}$ are conditionally dependent, once $\mathbf{x}$ is observed (Husmeier, 2005). Intuitively, we want some of the latent variables to represent relative spatial concepts—e.g. `left of` the cube in the image. For such concepts, under the same robot trajectory $\mathbf{x}$ we should get different latent codes, given different contexts $\mathbf{i}$. The omission of $\mathbf{y}$ from $q_\phi$ means that we might not be able to fully close the gap between the data distribution and its VLB, leading to a less efficient optimisation procedure (due to higher variance). That is mainly owing to the fact that we use weak supervision—only for some demonstrations do we have labels $\mathbf{y}$. The derivation of and commentary on the optimised VLB are provided in the Supplementary Materials.

$$\log p(\mathbf{x}, \mathbf{y}|\mathbf{i}) \geq \mathbb{E}_{q_\phi(\mathbf{c}|\mathbf{x}, \mathbf{i})}(\log p_\theta(\mathbf{x}, \mathbf{y}|\mathbf{c}, \mathbf{i})) - D_{KL}(q_\phi(\mathbf{c}|\mathbf{x}, \mathbf{i})||p(\mathbf{c})) \quad (4) \qquad q_\phi(\mathbf{c}|\mathbf{x}, \mathbf{i}) = \mathcal{N}(\mathbf{c}|\boldsymbol{\mu}_\phi(\mathbf{x}, \mathbf{i}), \boldsymbol{\sigma}_\phi(\mathbf{x}, \mathbf{i})) \quad (5)$$

## 3 Weak Labelling and User Specification

**Interpretability through Weak Labels** Even though the latent variables are constrained to be useful both for the tasks of trajectory and label generation, nothing forces them to carry the intuitive absolute and relative notions described in Section 2 in a factorised fashion—e.g. for dabbing, $c_1$ to correspond to the effort applied when pressing against the table, $c_2$ to correspond to how quickly the robot presses against the table, etc. Under an optimised model, those notions can be captured in $\mathbf{c}$ but with no guarantee that they will be represented in an **interpretable** way— i.e. $\mathbf{c}$ needs to be

**disentangled**. We achieve disentanglement in the latent space by establishing a one-to-one corre-spondence between a subset of the latent axes—$\mathbf{c}_s, \mathbf{c}_e$—and the concept groups in $\mathcal{G}$ (Hristov et al., 2018). $\mathbf{c}_s$ is optimised to represent spatial notions, $\mathbf{c}_e$—effort-related and temporal notions. The rest of the latent dimensions—$\mathbf{c}_u$—encode features in the data which don't align with the semantics of the labels and their concept groups but are still necessary for good trajectory reconstruction (Figure 2). Under these assumptions the label likelihood from Eq. 3 becomes:

$$p_\theta(\mathbf{y}|\{\mathbf{c}_s, \mathbf{c}_e\}) = \prod_j^{|\mathcal{G}|} \mathbf{1}_{\{y_j \neq \emptyset\}} \text{Cat}(y_j | \boldsymbol{\pi}_j(c_j)) \tag{6}$$

Labelling is **weak/restricted** since each demonstration is given just a single label from a single concept group. This is meant to represent a more natural LfD scenario, where the main task of the expert is still to perform a number of demonstrations, rather than to exhaustively annotate each demonstration with relevant labels from all groups. For example, a demonstrator might say *"this is how you press softly"* and proceed to perform the demonstration (as is common, say, in an instruc-tional video). The demonstrated behaviour might also be `slow` and `short` but the demonstrator may not have been explicitly thinking about these notions, hence may not have labelled accordingly. The missing label values are incorporated with an indicator function in Eq. 6.

**Condition on User Specification**  Apart from being used in the optimisation objective, the weak labels are also used in an additional conditioning step at test time. The generative process we have consists of first sampling values for $\mathbf{c}$ from its prior. These are then passed together with $\mathbf{i}$ through $p_\theta$ in order to get distributions over $\mathbf{x}$ and $\mathbf{y}$, from which we can sample. However, we are really interested in being able to incorporate the provided information about the labels $\mathbf{y}$ into the sampling of semantically aligned parts of $\mathbf{c}$. Specifically, we are interested in being able to generate robot behaviours which are consistent with chosen values for $\mathbf{y}$ (what we call a **user specification**). Post-optimising the $\theta$ and $\phi$ parameters, we choose to approximate the posterior over $\mathbf{c}$ for each label $l$ in each concept group $j$ with a set of Gaussian distributions $\mathcal{N}(\boldsymbol{\mu}_{jl}, \boldsymbol{\Sigma}_{jl})$. We use Maximum Likelihood Estimation for $\boldsymbol{\mu}_{jl}$ and $\boldsymbol{\Sigma}_{jl}$ over a set of samples from $\mathbf{c}$. The samples are weighted with the corresponding label likelihood $p_\theta(y_j = l | \boldsymbol{\pi}_j(c_j))$. As a result, the process of generating a trajectory $\mathbf{x} \sim p_\theta(\mathbf{x}|\mathbf{i}, \mathbf{c})$ can be additionally conditioned on an optional user specification $\mathbf{y} = \{y_1, \ldots, y_{|\mathcal{G}|}\}$, through $\mathbf{c}$, in a single-label fashion (Eq. 7) or a compositional fashion (Eq. 8). Pseudo-code for the generation of trajectories for both types of conditioning is provided in the Supplementary Materials.

$$\mathbf{c} \sim p(\mathbf{c}|y_j = l) = \mathcal{N}(\mathbf{c}|\boldsymbol{\mu}_{jl}, \boldsymbol{\Sigma}_{jl}) \tag{7}$$

$$\forall j \in \{1, \ldots, |\mathbf{c}|\}: c_j \sim p(c_j|y_j = l) = \begin{cases} \text{prior,} & \text{if } y_j = \emptyset \text{ or } j > |\mathcal{G}| \\ \mathcal{N}(\mathbf{c}|\boldsymbol{\mu}_{jl}, \boldsymbol{\Sigma}_{jl})_{(j)}, & \text{otherwise} \end{cases} \tag{8}$$

## 4 METHODOLOGY

In terms of a concrete model architecture, our model is a Conditional Variational Autoencoder (CVAE) (Sohn et al., 2015) with an encoding network $q_\phi$ and a decoding network $p_\theta$. The pa-rameters of both networks are optimised jointly using methods for amortised variational inference and a stochastic gradient variational bayes estimation (Kingma & Welling, 2013).[2]

**Encoding & Decoding Networks**  Due to the diversity of modalities that we want to use, $q_\phi$ is implemented as a combination of 2D convolutions (for the image input), 1D convolutions (for the trajectory input) and an MLP that brings the output of the previous two modules to the common concept manifold $\mathbf{c}$. For the decoding network, we implement a Temporal Convolutional Network (TCN) (Bai et al., 2018) which operates over the concatenation $\mathbf{h}$ of a single concept embedding $\mathbf{c}$ and a single image encoding $\mathbf{i}$. However, the TCN takes a sequence of length $T$ and transforms it to another sequence of length $T$, by passing it through a series of dilated convolutions, while $\mathbf{h}$ is equivalent to a sequence of length 1. Thus, we tile the concatenated vector $T$ times and to each

---

[2]The models are implemented in PyTorch (Adam et al., 2017) and optimised using the Adam optimiser (Kingma & Ba, 2014)

instance of that vector $\mathbf{h}_i, i \in \{1, \ldots, T\}$, we attach a time dimension $t_i = \frac{i}{T}$. This broadcasting technique has previously shown to achieve better disentanglement in the latent space both on visual (Watters et al., 2019) and time-dependent data (Noseworthy et al., 2019). We only set up the mean $\boldsymbol{\mu}_\theta$ of the predicted distribution over $\mathbf{x}$ to be a non-linear function of $[\mathbf{h}; \mathbf{t}]$ while $\boldsymbol{\sigma}$ is fixed, for better numerical stability during training. This allows us to use the $L2$ distance—Mean Squared Error (MSE)—between the inferred trajectories $\boldsymbol{\mu}_\theta$, acting as a reconstruction, and the ground truth ones.

**Label Predictors** For each concept group $g_j$ we take values from the corresponding latent axis $c_j$ and pass it through a single linear layer, with a softmax activation function, to predict a probability vector $\boldsymbol{\pi}_j$. Maximising the label likelihood is realised as a Softmax-Cross Entropy (SCE) term in the loss function. Optimising this term gives us a better guarantee that some of the latent axes will be semantically aligned with the notions in the label groups. The fact that some labels might be missing is accounted for with an indicator function which calculates $\nabla\theta, \nabla\phi$ with respect to the SCE, only if $y_j$ is present.

**Weighted Loss** The full loss function that we optimise is presented in Eq. 9. It is composed of three main terms—the trajectory MSE, equivalent to the negative trajectory log-likelihood, the label SCE, equivalent to the negative weak label log-likelihood and the KL divergence between the fit amortised posterior over $\mathbf{c}$ and its prior. The concrete values of the three coefficients—$\alpha, \beta, \gamma$—are discussed in the Supplementary Materials.

$$\min_{\theta,\phi,\mathbf{w}} \mathcal{L}(\mathbf{x}, \mathbf{y}, \mathbf{I}) = \beta D_{KL}(\underbrace{q_\phi(\mathbf{c}|\mathbf{x}, \mathbf{I})}_{\substack{\text{amortised} \\ \text{posterior}}} || \underbrace{p_\theta(\mathbf{c})}_{\text{prior}}) + \alpha \mathbb{E}_{q_\phi(\mathbf{c}|\mathbf{x}, \mathbf{I})}(\log \underbrace{p_\theta(\mathbf{x}|\mathbf{c}, \mathbf{I})}_{\text{MSE}}) + \gamma \sum_i^{|\mathcal{G}|} \mathbf{1}_{\{y_i \neq \emptyset\}} \underbrace{H(c_i \mathbf{w}_i^T, y_i)}_{\text{SCE}}$$
(9)

## 5 EXPERIMENTS

The models we experiment with include a Variational Autoencoder (Kingma & Welling, 2013) with a Gaussian prior (VAE), an Adversarial Autoencoder (Makhzani et al., 2015) with a Uniform prior (AAE) and a VAE with mixed continuous and categorical latent variables (GS). For the last model, the continuous variables have a Gaussian prior and the categorical variables utilise the Gumbel-Softmax reparametrisation (Jang et al., 2017). Each of the three models, is trained while utilising the weak labels—VAE-weak, AAE-weak, GS-weak—and with no label information—VAE, AAE, GS. Whether or not labels are used during training is controlled by setting $\gamma = 0$ in Eq. 9.

The setup used in the experiments consists of a PR2 robot, an HTC Vive controller and headset, and a tabletop with a single object on it—a red cube. Multiple pressing motions on the surface of the table are demonstrated by teleoperating the end effector of the robot. The pose of the end effector, with respect to the robot's torso, is mapped to be the same as the pose of the controller (held by the demonstrator) with respect to the headset (behind the demonstrator in Figure 4).

**Data** A total of 100 demonstrations were captured from the PR2, consisting of:

- Initial frame of each demonstration: 128x128-pixel Kinect2 image (RGB);
- 7-DoF joint angle positions;
- 7-DoF joint efforts;
- 5 groups of discrete weak labels;

Each demonstration has a single label attached to it. In total, we have 4 spatial symbols—where in the image, with respect to the red cube, do we dab—and 6 force-related symbols—how do we dab (Table 1). Figure 3 provides a graphical depiction demonstrating the differences between the different force-relate groupings qualitatively.

All trajectories are standardised to be of fixed length—the length of the longest trajectory (240 timesteps)—by padding them with the last value for each channel/joint. Additionally, both the joint posi-

| spatial (where) Image + Joint Angles | effort (how) Joint Efforts |
|:---:|:---:|
| **left & right** **front & behind** | **soft & hard** **short & long** **fast & slow** |

Table 1: All labels and modalities used in the demonstrations.

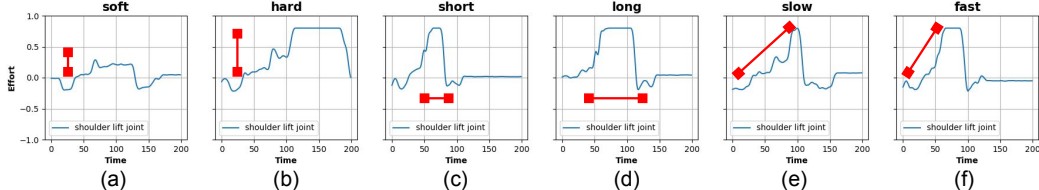

Figure 3: Example user-given labels across 6 different demonstrations. Depicted are effort trajectories from the training data for a single robot joint—`shoulder lift joint`—**(a)** and **(b)** designate the maximal exerted effort, **(c)** and **(d)** designate the length for which the maximal effort is maintained, **(e)** and **(f)** designate the time to reach the maximal effort (slope).

tions and efforts are augmented with Gaussian noise or by randomly sliding them leftwards (simulating an earlier start) and padding accordingly. The size of the total dataset after augmentation is 1000 demonstrations which are split according to a 90-10 training-validation split. The size of the latent space is chosen to be $|\mathbf{c}| = 8$. In the context of the problem formulation in Section 2, $\mathbf{c}_s = \{c_0, c_1\}, \mathbf{c}_e = \{c_2, c_3, c_4\}$ and $\mathbf{c}_u = \{c_5, c_6, c_7\}$.

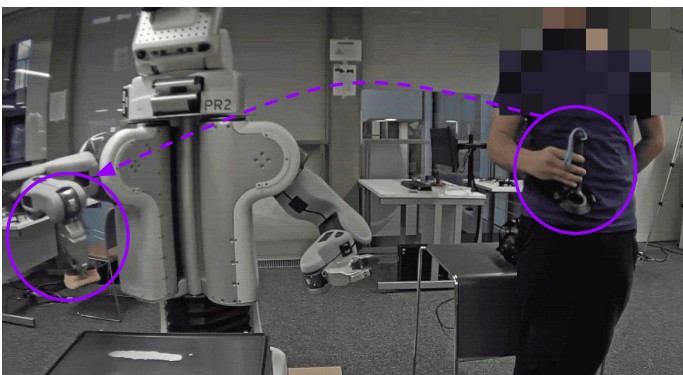

Figure 4: Physical setup for teleoperating a PR2 end-effector through an HTV Vive controller.

**Conditioning on a Single Label** We use the Gaussian distributions for each label $l$ in each concept group $j$—$\mathcal{N}(\boldsymbol{\mu}_{jl}, \boldsymbol{\Sigma}_{jl})$—to sample and generate trajectories that are consistent with the meaning of a particular label. For a set of 5 test images, which have not been seen during training, we draw 1000 samples of $\mathbf{c}$ per image for each of the 10 labels. For each label, we then judge whether the generated trajectories match the semantics of the corresponding label using a series of manually-designed heuristics, described in the Supplementary Materials. Average accuracy across all images and all samples is reported for each label. For the labels of a spatial nature, for each 7-dimensional (7 DoFs) robot state $x_i, \mathbf{x} = \{x_1, \ldots, x_T\}$, we compute the corresponding end-effector pose $p_i$ through the forward kinematics $\mathbf{K}$ of the robot. Using the camera model of the Kinect2 sensor, each pose $p_i$ is projected in the image we condition on. We report the Mean Absolute Error (MAE) in normalised pixel coordinates between the dab location and the landmark location (in pixel coordinates) for the misclassified trajectories. This gives us a sense of how far from the desired ones the generated samples are. The dab location is inferred as the point in the end-effector pose trajectory with the lowest $z$ coordinate. For example, if we have conditioned on the `left` label but some of the sampled trajectories result in a dab to the `right` of the cube in the image, we report *how far* from the true classification of `left` was the robot when it touched the table.

**Conditioning on a Composition of Labels** Since the concepts we encode in the latent space of the model should be independent of each other they can be composed together—e.g. a trajectory can be simultaneously `hard`, `long` and `fast`. We examine how well is this conceptual independence preserved in the latent space by generating trajectories conditioned on all possible combinations of labels regarding effort, speed and length of press—8 possible combinations in total. We still use the Gaussian distributions for each label in each concept group but only take the values along the relevant latent axis—Eq. 8. To fully close the loop, the trajectories which we sample from the model could further be executed on the physical robot through a hybrid position/force controller (Reibert, 1981). However, such an evaluation is beyond the scope of the paper.

## 6 RESULTS & DISCUSSION

Our experiments demonstrate that the models which utilise weak user labels within our proposed setup can more reliably generate behaviours consistent with a given spatial or force-related user specification, in contrast to baseline models which do not benefit from such information. Moreover, we find that continuous latent variables better capture the independent continuous notions which underlie the discrete user labels in comparison to discrete latent variables. Below we present quantitative results from the experiments with further qualitative results in the Supplementary Materials.

When conditioning on a single force label—Table 3—the best-performing models are VAE-weak and AAE-weak. In the worse-performing cases—VAE, GS—it would appear that the models suffer from partial mode collapse. For instance, in Table 3, penultimate row, the VAE can not represent `soft` trajectories, most samples are `hard`. Interestingly AAE and AAE-weak perform similarly well. We attribute that to the fact the Uniform prior used does not have a discrete number of modes to which the optimised posterior might potentially collapse. Simultaneously, regardless of their prior, none of the *weak* models suffers from mode collapse, since the label information helps for the corresponding notions to be more distinguishable in the latent space. All models have high accuracy, when conditioned on a spatial label—Table 2. However, GS-weak, VAE-weak and AAE-weak appear to be closer to the true conditioned-on concept when a bad trajectory is generated, as measured by the lower Mean Absolute Error of the image-projected end-effector positions.

| Model | left | | right | | front | | back | |
|---|---|---|---|---|---|---|---|---|
| | Acc | MAE | Acc | MAE | Acc | MAE | Acc | MAE |
| GS | 0.83 (0.09) | 0.052 | 0.76 (0.22) | 0.040 | 0.85 (0.09) | 0.039 | 0.90 (0.10) | 0.060 |
| GS-weak | 0.78 (0.16) | **0.034** | **0.92** (0.13) | **0.013** | **0.92** (0.09) | **0.030** | **0.94** (0.06) | **0.028** |
| AAE | 0.84 (0.12) | 0.043 | 0.88 (0.09) | 0.043 | 0.81 (0.15) | 0.039 | 0.97 (0.03) | 0.024 |
| AAE-weak | 0.84 (0.24) | **0.018** | **0.89** (0.12) | **0.039** | **0.83** (0.19) | **0.030** | **0.98** (0.04) | **0.022** |
| VAE | 0.80 (0.18) | 0.026 | 0.82 (0.19) | 0.031 | 0.86 (0.08) | 0.050 | 0.97 (0.05) | 0.029 |
| VAE-weak | **0.95** (0.07) | 0.031 | **0.87** (0.14) | **0.026** | **0.93** (0.10) | **0.027** | **0.99** (0.02) | **0.018** |

Table 2: Accuracy (mean and std deviation; higher mean is better) and MAE (lower is better) for sampled trajectories under all models, conditioned on a fixed spatial label. Bold numbers designate when the weak version of the model (utilising the weak user labels) outperforms the non-weak one.

| Model | soft | hard | short | long | slow | fast |
|---|---|---|---|---|---|---|
| GS | 0.26 (0.01) | 0.96 (0.01) | 0.95 (0.00) | 0.76 (0.01) | 0.92 (0.01) | 0.77 (0.01) |
| GS-weak | **0.84** (0.01) | **0.98** (0.00) | **0.98** (0.00) | 0.69 (0.01) | 0.88 (0.01) | 0.74 (0.02) |
| AAE | 0.98 (0.00) | 1.00 (0.00) | 1.00 (0.00) | 0.75 (0.01) | 0.93 (0.01) | 0.64 (0.01) |
| AAE-weak | 0.98 (0.00) | 0.99 (0.00) | 0.99 (0.00) | **0.97** (0.01) | 0.84 (0.01) | **0.84** (0.01) |
| VAE | 0.04 (0.01) | 0.95 (0.00) | 0.96 (0.01) | 0.59 (0.01) | 0.93 (0.01) | 0.52 (0.01) |
| VAE-weak | **0.92** (0.00) | **1.00** (0.00) | **1.00** (0.00) | **0.90** (0.01) | **0.95** (0.00) | **0.68** (0.01) |

Table 3: Accuracy (mean and std deviation) for sampled trajectories under all models, conditioned on a fixed effort label. Bold numbers designate when the weak version of the model (utilising the weak user labels) outperforms the non-weak one.

While single-label conditioning is possible under all models, it is only VAE-weak, AAE-weak and GS-weak which provide a reliable mechanism for *composing* labels from different independent concept groups. That is possible due to the explicit alignment of each concept and its labels with a single latent axis. Trajectory generation is done after conditioning on compositions of labels from the three independent concept groups designating effort—see Table 1 (right column). Specifically, the difference between the sampled latent values for `hard long slow` and `hard long fast`, for example, are only along the axis used to predict `slow` from `fast`—$c_4$ in this case. If the generated trajectories match their specification (high measured accuracy) then indeed the perturbed axis is aligned with the notion behind the perturbed labels. Results from this experiment are presented in Figure 5. The better-performing models are the ones with continuous latent variables. While the VAE-weak and AAE-weak perform equally well for most label combinations and do manage to

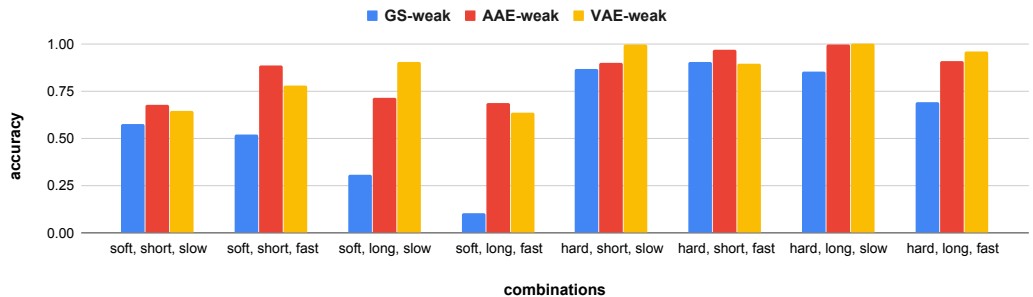

Figure 5: Accuracy comparison for generated trajectories conditioned on all compositions of binary labels from the three concept groups for effort, length and speed. The models with continuous latent spaces outperform the discrete one as the underlying concepts are continuous themselves.

encode independent concepts as such (along specific latent axes), the model with the mix of continuous and categorical latent variables (GS-weak) does not match their results. Taking the second row in Table 3 into account, this means that the categorical model (which has more constrained capacity than the VAE-weak and AAE-weak) can capture the data distribution (shows good single-label-conditioning reconstructions) but in an entangled fashion. Namely, going from `slow` to `fast` corresponds to perturbing more than a single latent axis simultaneously. This further supports our initial hypothesis that while discrete, the weak labels only provide a mechanism for shaping the latent representation of what are otherwise continuous phenomena.

## 7 POURING EXPERIMENT

An additional pouring task is demonstrated where the manner of execution varies spatially and in terms of how it is executed—see Table 4 and Figure 6. 20 demonstrations are captured for each label. The size of the latent space is kept to be $|\mathbf{c}| = 8$. In the context of the problem formulation in Section 2, $\mathbf{c}_s = \{c_0\}$, $\mathbf{c}_e = \{c_1, c_2\}$ and $\mathbf{c}_u = \{c_3, c_4, c_5, c_6, c_7\}$.

| red cup | blue cup | fully | partially | sideways | behind |
|---------|----------|-------|-----------|----------|--------|

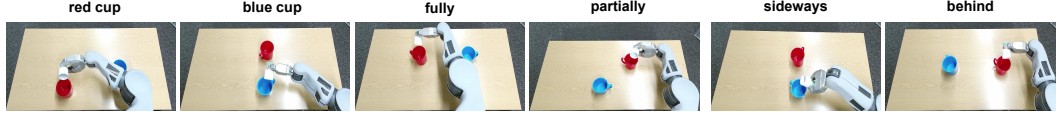

Figure 6: Example user-given labels across 6 different demonstrations for pouring.

When compared to the dabbing setup, this task presents a more significant challenge in identifying the switching latent dynamics for pouring in each cup (compared to the single red cube in the previous task, acting as a reference point for resolving spatial prepositions).

| spatial (where) | manner (how) |
|---|---|
| Image + Joint Angles | Joint Angles |
| **red cup & blue cup** | **behind & sideways** |
| | **fully & partially** |

Table 4: All labels and modalities used in the additional pouring demonstrations.

Interpolation experiments for the VAE-weak model are presented in Figure 7 for 4 test cup configurations. The first latent dimension of the model—$c_0$, optimised to predict in which cup to pour—is linearly interpolated in the range $[-2, 2]$. In all four rows, we can see the end effector projections in the image plane, through the robot kinematics and camera model, showing a gradual shift from initially pouring in the `red cup` to eventually pouring in the `blue cup`. The color evolution of each end-effector trajectory from cyan to magenta signifies time progression. Moreover, given the disentangled nature of the VAE-weak model, we also generate robot trajectories conditioned on specific values for the 3 latent axes ($c_0$, $c_1, c_2$), consistent with example label compositions—see Figure 8. For picking the right value for each axis we use the post-training-estimated Normal distributions for each label (consult Section 3). Additional qualitative results and sensitivity analysis of the model to the heterogeneous input modalities are presented in Appendix E and the supplementary website [3].

---

[3]https://sites.google.com/view/weak-label-lfd

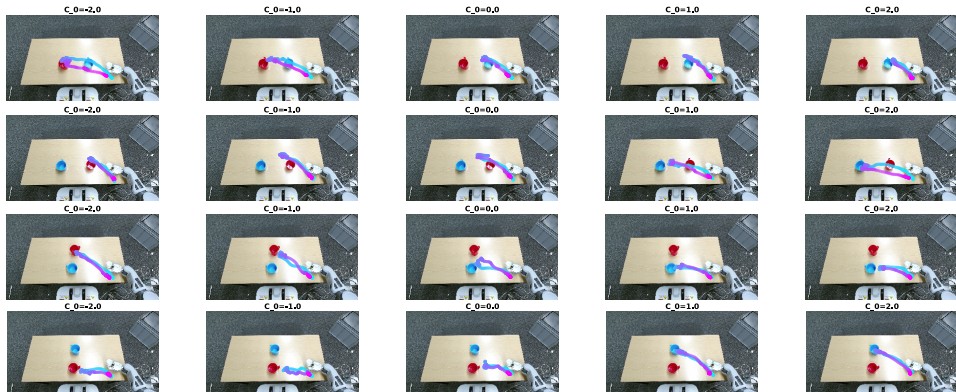

Figure 7: Linear interpolation from pouring in the `red cup` to pouring in the `blue cup`.

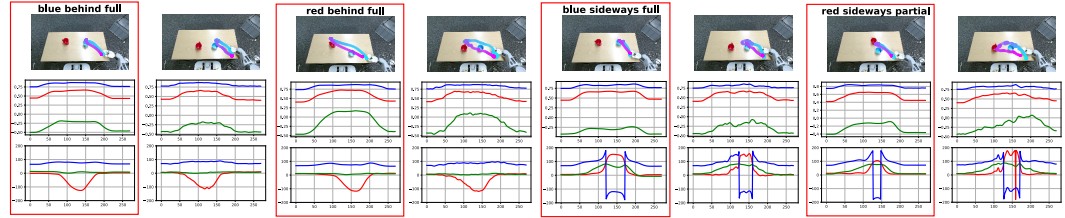

Figure 8: Qualitative comparison of demonstrated robot trajectories and sampled trajectories, generated after conditioning on a composition of labels. Each pair consists of a curated ground-truth demo, characterised by a particular label composition (red boxes) and a respective sample from the optimised model. Each sample plot consists of in-image projected end effector trajectories (top row) together with x/y/z (middle row) and roll/pitch/yaw (bottom row) end effector trajectories.

## 8    RELATED WORK

Methods that utilise high-parameter neural models to learn disentangled low-dimensional representation achieve that by either tuning the optimised loss function (Higgins et al., 2017; Chen et al., 2018), imposing crafted priors over the latent space (Chen et al., 2016) or using weak supervision (Hristov et al., 2019; Rasmus et al., 2015). While being able to produce interpretable manifolds, most of these approaches focus on modelling visual data and respectively visually-manifested concepts, with minor exceptions—e.g. modelling motion capture data (Ainsworth et al., 2018). In contrast, we solve a problem involving multiple heterogeneous modalities—vision, forces, joint angles. Moreover, in addition to separating factors of variation in the data, to achieve latent space interpretability, we also deal with the problem of entanglement across modalities—i.e. which factors relate to which data modalities.

Dynamic Movement Primitives (Schaal, 2006) and Probabilistic Movement Primitives (Paraschos et al., 2013) are commonly used to model dynamical systems as 'point-mass attractors with a non-linear forcing term'. However, the resulting control parameters can be high-dimensional and un-intuitive for end-user manipulation. Performing dimensionality reduction (Colomé et al., 2014) or imposing hierarchical priors (Rueckert et al., 2015) are both ideas seen in the literature as a means of making the high-dimensional parameter space more meaningful to the human demonstrator. These have the advantage of yielding a clearer idea of how changes in the optimised parameters result in deterministic changes in the generated behaviours. However, most of these approaches limit themselves to end-effector trajectories, or – at most – making use of visual input (Kober et al., 2008).

Noseworthy et al. (2019) explore disentangling task parameters from the manner of execution parameters, in the context of pouring. They utilise an adversarial training regime, which facilitates better separation of the two types of parameters in the latent space. However, it is assumed that the task parameters are known *a priori*. Interpretation of the learned codes is achieved post-training by perturbing latent axis values and qualitatively inspecting generated trajectories (standard evaluation technique for models which are trained unsupervised). We argue that through the use of weak supervision through discrete labels we have finer control over the 'meaning' of latent dimensions.

## 9 CONCLUSION

We recontextualise the problem of interpretable multi-modal LfD as a problem of formulating a conditional probabilistic model. In the example task of table-top dabbing, we utilise weak discrete labels from the demonstrator to represent abstract notions, manifested in a captured multi-modal robot dataset. Through the use of high-capacity neural models and methods from deep variational inference, we show how to align some of the latent variables of the formulated probabilistic model with high-level notions implicit in the demonstrations.

### ACKNOWLEDGMENTS

This work is partly supported by funding from the Turing Institute, as part of the Safe AI for surgical assistance project. Additionally, we thank the members of the Edinburgh Robust Autonomy and Decisions Group for their feedback and support.

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

## A  VARIATIONAL LOWER BOUND DERIVATION

For completeness we include a derivation of the Variational Lower Bound of the data likelihood. The VLB is used as an optimisation objective—Eq. 9. Jensen's inequality becomes an equality when the amortised posterior $q(\mathbf{c}|\mathbf{x},\mathbf{i})$ matches exactly the true posterior $p(\mathbf{c}|\mathbf{x},\mathbf{i},\mathbf{y})$. The posterior we optimise is not conditioned on $\mathbf{y}$, potentially meaning that we might never close the gap between the $\log$ of the data distribution and the VLB, measured by $D_{KL}(q(\mathbf{c}|\mathbf{x},\mathbf{i})||p(\mathbf{c}|\mathbf{x},\mathbf{i},\mathbf{y}))$. However, maximising the VLB still goes in the direction of maximising the data distribution.

$$
\begin{aligned}
\log p(\mathbf{x},\mathbf{y}|\mathbf{i}) &= \log \int p(\mathbf{x},\mathbf{y}|\mathbf{i},\mathbf{c})p(\mathbf{c})d\mathbf{c} \\
&= \log \int p(\mathbf{x}|\mathbf{i},\mathbf{c})p(\mathbf{y}|\mathbf{c})p(\mathbf{c})d\mathbf{c} \\
&= \log \int \frac{q(\mathbf{c}|\mathbf{x},\mathbf{i})}{q(\mathbf{c}|\mathbf{x},\mathbf{i})}p(\mathbf{x}|\mathbf{i},\mathbf{c})p(\mathbf{y}|\mathbf{c})p(\mathbf{c})d\mathbf{c} \\
&= \log \mathbb{E}_{q(\mathbf{c}|\mathbf{x},\mathbf{i})} \frac{p(\mathbf{x}|\mathbf{i},\mathbf{c})p(\mathbf{y}|\mathbf{c})p(\mathbf{c})}{q(\mathbf{c}|\mathbf{x},\mathbf{i})} \\
&\geq \mathbb{E}_{q(\mathbf{c}|\mathbf{x},\mathbf{i})} \log \frac{p(\mathbf{x}|\mathbf{i},\mathbf{c})p(\mathbf{y}|\mathbf{c})p(\mathbf{c})}{q(\mathbf{c}|\mathbf{x},\mathbf{i})} \qquad\qquad \text{[Jensen]} \\
&= \mathbb{E}_{q(\mathbf{c}|\mathbf{x},\mathbf{i})} \log p(\mathbf{x}|\mathbf{i},\mathbf{c}) + \mathbb{E}_{q(\mathbf{c}|\mathbf{x},\mathbf{i})} \log p(\mathbf{y}|\mathbf{c}) - \mathbb{E}_{q(\mathbf{c}|\mathbf{x},\mathbf{i})} \log \frac{q(\mathbf{c}|\mathbf{x},\mathbf{i})}{p(\mathbf{c})} \\
&= \mathbb{E}_{q(\mathbf{c}|\mathbf{x},\mathbf{i})} \log p(\mathbf{x}|\mathbf{i},\mathbf{c}) + \mathbb{E}_{q(\mathbf{c}|\mathbf{x},\mathbf{i})} \log p(\mathbf{y}|\mathbf{c}) - D_{KL}(q(\mathbf{c}|\mathbf{x},\mathbf{i})||p(\mathbf{c})) \\
&= -(D_{KL}(q(\mathbf{c}|\mathbf{x},\mathbf{i})||p(\mathbf{c})) - \mathbb{E}_{q(\mathbf{c}|\mathbf{x},\mathbf{i})} \log p(\mathbf{x}|\mathbf{i},\mathbf{c}) - \mathbb{E}_{q(\mathbf{c}|\mathbf{x},\mathbf{i})} \log p(\mathbf{y}|\mathbf{c})) \\
&= -(\mathcal{L})
\end{aligned}
\tag{10}
$$

## B  LABEL CONDITIONING & EVALUATION HEURISTICS

Algorithms 1 and 2 provide pseudo-code for the trajectory generation procedures described in Section 3. The main difference can be summarised as following—when the generative process is conditioned on a single label $l$, all 8 dimensions of the latent samples are drawn from a single 8-dimensional Gaussian distribution associated with that label and its concept group—line 2 in Algorithm 1. However, when the process is conditioned on a composition of labels from different groups—a user specification—the sampling procedure differs as $\mathbf{c}$ is iteratively built. For each latent axis $c_j$, if a label is specified for the concept group aligned with $c_j$, only the values along the $j$-th axis are taken of a sample from the corresponding 8-dimensional Gaussian. If no label is given, $c_j$ is sampled from a Standard Normal distribution $\mathcal{N}(0,1)$ (or the respective prior of choice.

Same applies for latent dimensions which have not been aligned with any concept groups, since $|\mathbf{c}| > |\mathcal{G}|$—lines 2 to 9 in Algorithm 2.

---

**Algorithm 1:** Trajectory generation conditioned on a single label

---

**Input:** Visual Scene image $\mathbf{I}$
**Input:** Single user label $y$ from concept group $g$
**Input:** Image encoder, part of $p_\theta$
**Input:** Decoder network $q_\phi$
**Input:** set of Gaussian distribution for each label in each concept group:
$\quad\quad K = \{\{\mathcal{N}(\mu_{jl}, \Sigma_{jl})\}, j \in \{1, \ldots, |\mathcal{G}|\}, l \in \{1, \ldots, |g_j|\}$
**Output:** Generated trajectory $\hat{\mathbf{x}}$ of time length $T$

1 encode image input $\mathbf{i} \leftarrow p_\theta(\mathbf{I})$;
2 sample latent values $\mathbf{c} \sim \mathcal{N}(\mathbf{c}|\boldsymbol{\mu}_{jl}, \boldsymbol{\Sigma}_{jl})$, such that $j = g, l = y$;
3 concatenate $\mathbf{c}$ and $\mathbf{i}$: $\mathbf{h} \leftarrow [\mathbf{c}; \mathbf{i}]$;
4 tile $\mathbf{h}$ $T$ times;
5 attach a last time dimension $\mathbf{h} = [\mathbf{h}; \mathbf{t}], \mathbf{t} = \{t_1, \ldots, t_T\}, t_i = \frac{i}{T}$;
6 generate trajectory $\hat{\mathbf{x}} \leftarrow q_\phi(\mathbf{x}|\mathbf{h})$;
7 return $\hat{\mathbf{x}}$;

---

**Algorithm 2:** Trajectory generation conditioned on a composition of labels

---

**Input:** Visual Scene image $\mathbf{I}$
**Input:** User label specification $\mathbf{y} = \{y_1, \ldots, y_{|\mathcal{G}|}\}$, potentially one label for each concept group
**Input:** Image encoder, part of $p_\theta$
**Input:** Decoder network $q_\phi$
**Input:** set of Gaussian distribution for each label in each concept group:
$\quad\quad K = \{\mathcal{N}(\mathbf{c}|\boldsymbol{\mu}_{jl}, \boldsymbol{\Sigma}_{jl})\}, j \in \{1, \ldots, |\mathcal{G}|\}, l \in \{1, \ldots, |g_j|\}$
**Output:** Generated trajectory $\hat{\mathbf{x}}$ of time length $T$

1 encode image input $\mathbf{i} \leftarrow p_\theta(\mathbf{I})$;
2 init $\mathbf{c} = []$;
3 **for** *each latent dimension $c_j \in \mathbf{c}$* **do**
4 $\quad$ **if** $y_j \neq \emptyset$ *and* $j <= |\mathcal{G}|$ **then**
5 $\quad\quad$ sample $\mathbf{w} \sim \mathcal{N}(\mathbf{w}|\boldsymbol{\mu}_{jl}, \boldsymbol{\Sigma}_{jl})$, such that, $l = y_j$;
6 $\quad\quad$ only take values along the $j$-th dimension: append $\mathbf{w}[j]$ to $\mathbf{c}$;
7 $\quad$ **else**
8 $\quad\quad$ sample $w \sim \mathcal{N}(w|0, 1)$;
9 $\quad\quad$ append $w$ to $\mathbf{c}$;
10 concatenate $\mathbf{c}$ and $\mathbf{i}$: $\mathbf{h} \leftarrow [\mathbf{c}; \mathbf{i}]$;
11 tile $\mathbf{h}$ $T$ times;
12 attach a last time dimension $\mathbf{h} = [\mathbf{h}; \mathbf{t}], \mathbf{t} = \{t_1, \ldots, t_T\}, t_i = \frac{i}{T}$;
13 generate trajectory $\hat{\mathbf{x}} \leftarrow q_\phi(\mathbf{x}|\mathbf{h})$;
14 return $\hat{\mathbf{x}}$;

---

The hand-designed heuristics used for reporting accuracy on the the generated trajectories have concrete semantics and values which have been chosen after closely inspecting the captured dataset, the given user labels and the robot configuration during the demonstrations. :

- spatial labels—`left`, `right`, `front`, `behind`—the location of the red object in the scene is extracted using standard computer vision techniques based on color segmentation. The position of the cube is inferred as the center of mass of all red pixels detected. From the overall end-effector trajectory, derived through the robot forward kinematics, the pressing location is chosen as the end-effector position with the lowest $z$ coordinate. That position is then projected into the image plane in order to determine where with respect to the detected object position did the robot touch the table.

- `soft` vs `hard` trajectories - the generated trajectories have normalised values in the range $[-1, 1]$. A `soft` trajectory is considered as such if specific joint efforts are below a particular threshold. More specifically:

- – `shoulder lift joint` efforts are below 0.5.
- – `upper arm roll joint` effort are above -0.5.

- `short` vs `long` trajectories - the generated trajectories have length of $T = 240$. A `short` trajectory is considered as such if efforts in the range of [max_effort-0.2, max_effort] are maintained for more than 50 time steps. Otherwise the trajectory is deemed `long`. The heuristic operates over the following joints:
  - – shoulder lift joint efforts
  - – upper arm roll joint efforts

- `slow` vs `fast` trajectories - a `slow` trajectory is considered as such if the time it takes to reach maximal joint effort is more than 80 time steps. Otherwise the trajectory is deemed `fast`. The heuristic operates over the following joints:
  - – shoulder lift joint efforts
  - – upper arm roll joint efforts

For resolving robot joint names, please consult Figure 9 below:

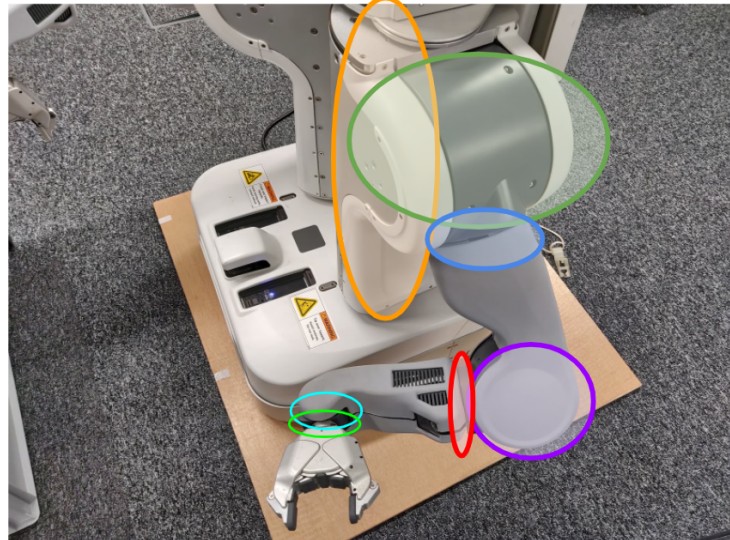

- **shoulder panning joint**
- **shoulder lift joint**
- **upper arm roll joint**
- **elbow flex joint**
- **forearm roll joint**

Figure 9: Color-coded robot arm joints and their corresponding names.

## C  ARCHITECTURE DETAILS

The model architectures are implemented in the PyTorch framework[4] and is visually presented in Figure 10. The image encoder network takes as input a single RGB 128x128 pixel image. The trajectory encoder takes a single 14-dimensional, 240 time-step-long trajectory. Their outputs both feed into an MLP network which, after a series of nonlinear transformations, outputs parameters of a distribution over the latent space **c**. Through the reparametrisation trick Kingma & Welling (2013) we sample values for **c**. Through a residual connection, the output of the image encoder is concatenated to the sampled latent values. The resultant vector is tiled 240 times, extended with a *time* dimension, and fed into a TCN decoder network to produce reconstructions for the original 14-dimensional input trajectories.

Across all experiments, training is performed for a fixed number of 100 epochs using a batch size of 8. The dimensionality of the latent space $|\mathbf{c}| = 8$ across all experiments. The Adam optimizer (Kingma & Ba, 2014) is used through the learning process with the following values for its parameters—($learning rate = 0.001, \beta1 = 0.9, \beta2 = 0.999, eps = 1e-08, weight decay rate = 0, amsgrad = False$).

---

[4]https://pytorch.org/docs/stable/index.html

| Image Encoder |
| :---: |
| FC (4) $\mathbf{i}$ |
| FC (256) |
| 2D Conv (k=3, p=1, c=64) |
| 2D Conv (k=3, p=1, c=64) |
| 2D Conv (k=3, p=1, c=64) |
| 2D Conv (k=3, p=1, c=32) |
| 2D Conv (k=3, p=1, c=32) |
| Input Image $\mathbf{I}$ [128 x 128 x 3] |

(a) Image Encoder

| Trajectory Encoder |
| :---: |
| FC (32) $\boldsymbol{\tau}$ |
| FC (256) |
| 1D Conv (k=7, p=3); 1D Conv (k=5, p=2); 1D Conv (k=3, p=1) [c=20] |
| Input Trajectory $\mathbf{x}$ [240 x 14] |

(b) Trajectory Encoder

| MLP |
| :---: |
| FC (2x8) $\mu, \log(\sigma)$ |
| FC (32) |
| FC (32) |
| Concatenated $[\mathbf{i}; \boldsymbol{\tau}]$[1 x 36] |

(c) MLP

| TCN Decoder |
| :---: |
| Temporal Block (dilation=4, k=5, c=14) |
| Temporal Block (dilation=2, k=5, c=20) |
| Temporal Block (dilation=1, k=5, c=20) |
| append time channel $\mathbf{t}$ |
| tile (240, 12) |
| Concatenated $[\mathbf{i}; \mathbf{c}]$[1 x 12] |

(d) TCN Decoder

Table 5: Network architectures used for the reported models. (a) is a 2D convolutional network, (b) is a 1D convolutional network, (c) is a fully-connected MLP network, (d) is a Temporal Convolution Network, made of stacked temporal blocks and dilated convolutions, described in Bai et al. (2018)

For all experiments, the values (unless when set to 0) for the three coefficients from Equation 9 are:

- $\alpha = 1, \beta = 0.1, \gamma = 10$

The values are chosen empirically in a manner such that all the loss terms have similar magnitude and thus none of them overwhelms the gradient updates while training the full model.

## D  QUALITATIVE EVALUATION FOR DABBING

Figures 11 to 15 present the qualitative results from perturbing in a controlled way the latent space optimised under the VAE-weak model. For figure the first row of plots represents the generated joint angle positions for each of the 7 joints, for each of the 5 drawn samples, the second row—the generated joint efforts—and the third row the corresponding end-effector positions, from the forward robot kinematics, projected in the image plane. As we can see, perturbing $c_0$ and $c_1$ corresponds to clear and consistent movement of the end effector (through the generated joint position profiles) while there is not much change in the generated effort profiles—Figures 11 and 12. Simultaneously, we observe the opposite effect in other 3 Figures. Perturbations in $c_2$ correspond solely changes in the output efforts in the shoulder lift and upper arm rolls joints—Figure 13 (i) and (j). Perturbations in $c_3$ correspond to changes in length for which the joint efforts are exerted but does not change their magnitude or where they are applied on the table—Figure 14. And finally, perturbations in $c_4$ correspond to changes on how quickly the maximum efforts are reached - most noticeable in the changing slope of the shoulder lift and upper arm rolls joints—Figure 15 (i) and (j). As a result, we can conclude that the latent dimensions, which we aligned with the semantically grouped weak labels, have indeed captured the notions and concepts which underlie the labels.

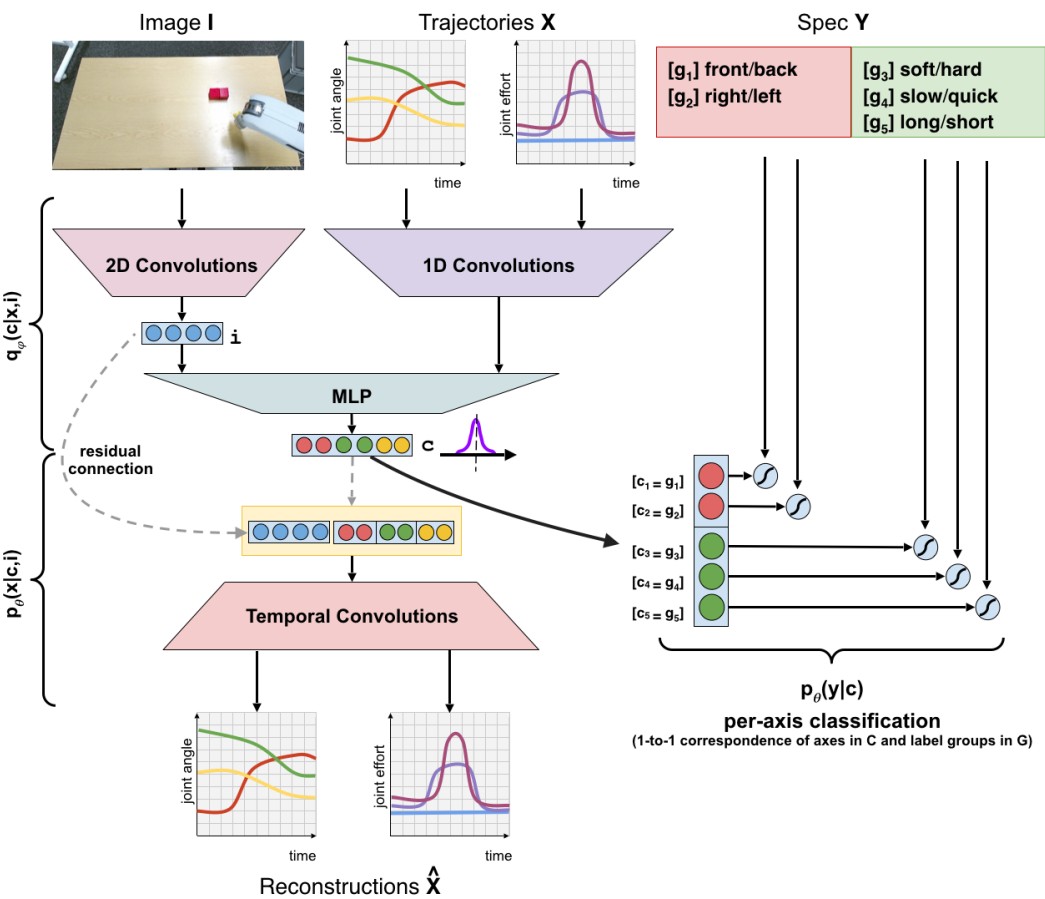

Figure 10: Visual schematic of the overall model. Multiple different modalities are encoded to a common latent space $\mathbf{c}$ which is used both for reconstructing trajectories $\mathbf{x}$ and predicting weak labels $\mathbf{y}$, coming from the demonstrator.

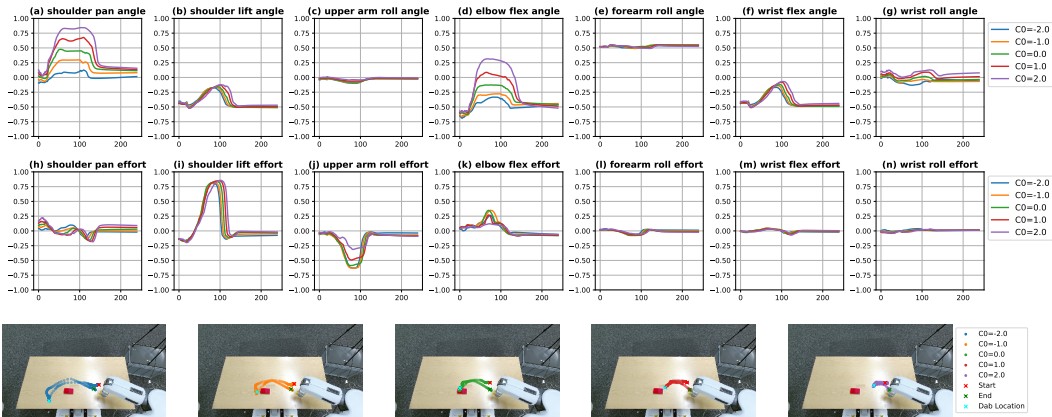

Figure 11: Linearly interpolate $c_0$ ($\equiv$ from dabbing to the `left` to dabbing to the `right` of the visual landmark in the scene.)

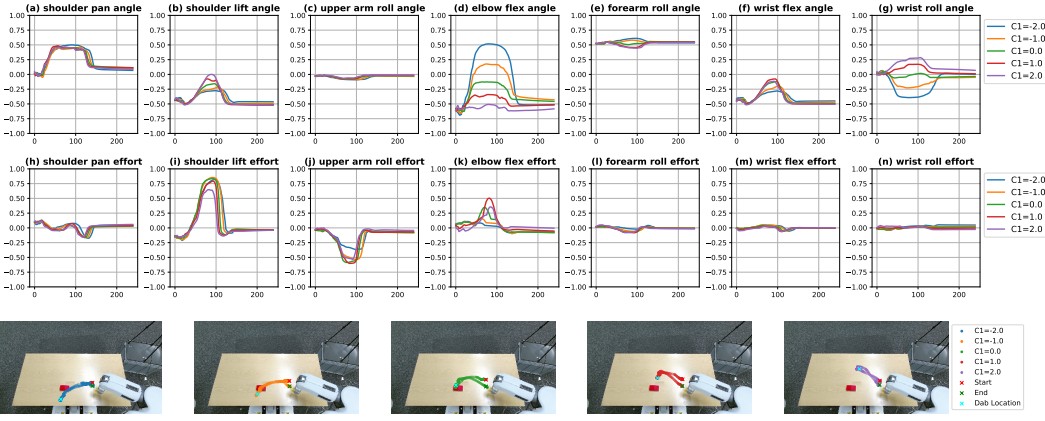

Figure 12: Linearly interpolate $c_1$ ($\equiv$ from dab to the `front` to dab to the `back` of the visual landmark in the scene.)

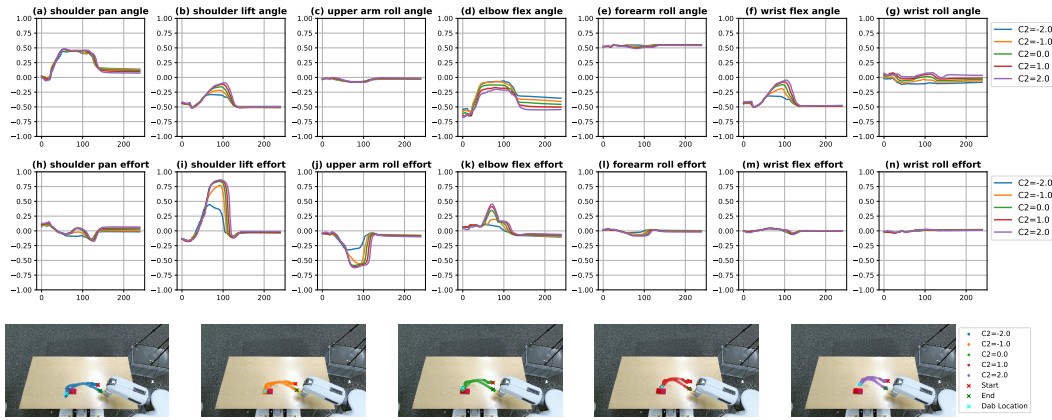

Figure 13: Linearly interpolate $c_2$ ($\equiv$ from dab `hardly` to `softly`)

## E MODEL SENSITIVITY ANALYSIS TO DIFFERENT DATA MODALITIES

As an exploration step towards analysing to what extend the auxiliary loss terms guide the model to pay attention to the different data modalities it is exposed to, we train the VAE and VAE-weak models on the demonstrations for pouring in the `red cup` and `blue cup`. The first latent dimension of the VAE-weak model is optimised to represent the conceptual variation of pouring in one of the two cups. Post-training, the values of the latent vectors **c** for both models are fixed and trajectory samples are drawn as the models are conditioned on four different test scenes, representing different cup configurations. Figure 16 demonstrates that, averaged across the different configurations, the VAE-weak model has greater variations in the sampled trajectories for all 7 joints of the robot arm when compared to the VAE one. This suggests that unless utilised, the input image modality might be partly ignored—the VAE does not have any auxiliary tasks to force it to use the scene image input, which can result in it just auto-encoding the trajectory data through **c**. In contrast, predicting where the robot pours in the scene at an abstract level is both a function of the joint angle trajectory and the image inputs—utilised through the classification loss of VAE-weak. Thus, for the same specification (fixed **c**), conditioning on different scenes leads to different joint trajectories.

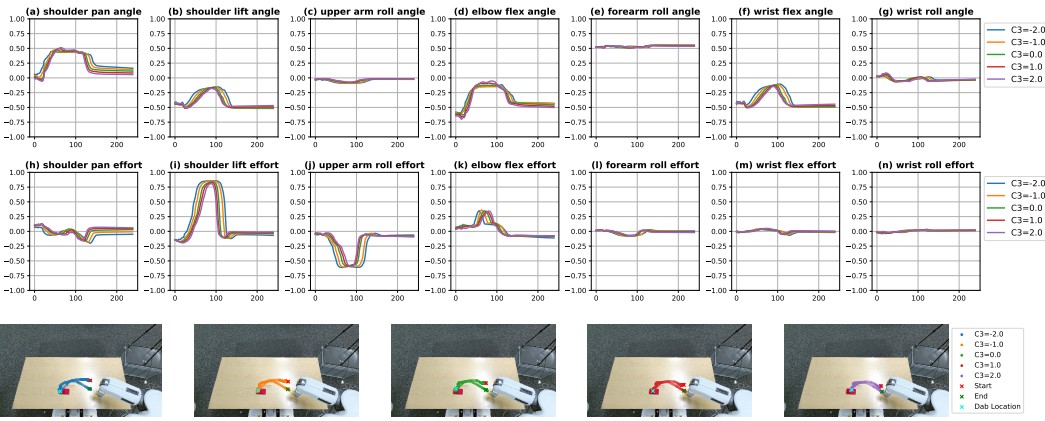

Figure 14: Linearly interpolate $c_3$ ($\equiv$ from dabbing for a `long` period of time to dabbing for a `short` period of time)

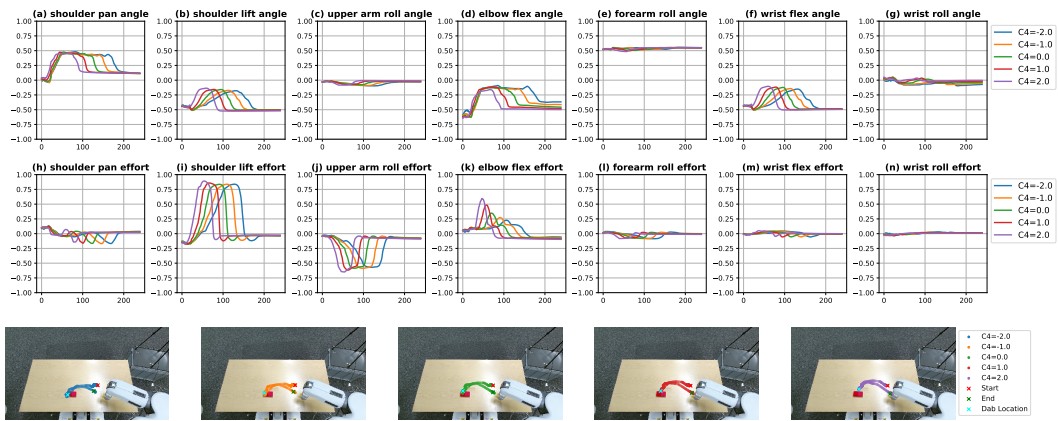

Figure 15: Linearly interpolate $c_4$ ($\equiv$ from dabbing `quickly` to dabbing `slowly`)

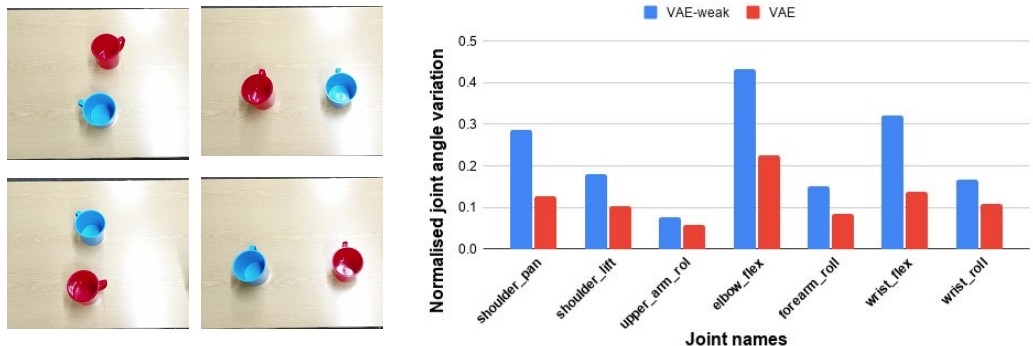

Figure 16: Four test cup configurations (**left**) and mean normalised joint angle variation for sampled trajectories conditioned on them (**right**).

