# OpenReview forum: "Learning from Demonstration with Weakly Supervised Disentanglement"
_ICLR.cc/2021/Conference — ICLR 2021 Poster_

### Official Review · AnonReviewer1 · 2020-10-28
**The paper needs real robot experiments**

**Rating:** 5
**Confidence:** 4

**Review:**

== SUMMARY ==

This is a well-written paper that discusses how to learn disentangled representations for the learning from demonstrations (LfD) task in robotics. It is shown that using weak-supervision on top of unsupervised learning frameworks (that use the variational autoencoder for instance) can work well in this case. These disentangled factors of variation in the data are shown to correspond well to the 'abstract concepts' of the human demonstrations. This is shown in the example of the PR2 robot dabbing demonstrations, including visual data as well robot trajectories.

== QUALITY & CLARITY ==

The paper is written very well, in fact most papers contain a lot of spelling mistakes but this paper was a joy to read in this regard :) The concepts are also explained clearly. However I would have expected better and more detailed coverage of related & past work.

== ORIGINALITY & SIGNIFICANCE ==

Unfortunately, I think the paper suffers from lack of originality, at least with respect to ML. From a robotics point of view, I would have accepted it as a very good application paper, if the authors had also presented real-robot results that show the learned model in action (generating actual trajectories for the robot). The authors however show only the prediction performance for (held-out) test demonstrations.

== VERDICT ==

I would like to thank the authors for a very well written paper. Overall, I think the paper needs some improvements such that it can be accepted in a revised version or most likely, in another conference. The paper needs to present real robot results and cover related work in more detail. Comparing the method to related work (DMP, or ProMP-variants, or any other competitive method) in the real-robot experiments would also be crucial.

=== NOTES & SOME MINOR COMMENTS ===

* No need to mention the link in the abstract

* "For example, the concept of pressing softly against a surface
manifests itselfin a data stream associated with the 7 DoF real-valued
space of joint efforts, spread across multipletime steps." -> For the PR2 robot? Either remove 7 DoF or add for which robot.

* "However, the essence of what differentiates one type of soft press from another nearby concept can be summarised conceptually using a lower dimensional abstract space" -> Nearby concept sounds vague, please be more explicit or give
examples.

* In Figure 2, the image encoding 'i' does not affect y. Why?

* Related Work: ProMPs are not represented as dynamical systems

* "To fully close the loop, the trajectories which we sample from the
model could further be executed on the physical robot through a hybrid
position/force controller (Reibert,1981). However, such evaluation is
beyond the scope of the paper." -> I don't think so. The real proof of concept is the actual robot
experiments!  Without real robot experiments to show how the generated
conditioned trajectories actually perform, in my opinion the paper is
an application paper without any significant ML contributions. The
paper as of now I fear only confirms the fact already acknowledged in
previous ML papers, that disentanglement can be achieved through
(semi)supervised learning [see Locatello et al. and papers citing
this work)

* equivallent -> equivalent

* force-relate -> force-related

---

> ### Author Response · Authors · 2020-11-24
> **Response to reviewer 1**
>
> We thank the reviewer for their comments and feedback.
>
> **Re. coverage of related & past work:** Within the constraints of tight page limits, and because we have added a new Section 7, we are restricted in how much we can survey. We are happy to provide further pointers to existing literature, in a potential camera-ready submission, given the permitted extra page. We would be grateful for specific pointers to references that are essential.
>
> **Re. Figure 2:** Figure 2 depicts the generative process for generating x and y. Some of the latent variables c are meant to encode continuous versions of the discrete, spatially-relative phenomena, described by y. Thus, knowing only the values of c is sufficient to infer y, regardless of the scene image, since both c_0 and y_0 are relative in nature. E.g. we could learn that c_0 being normally-distributed around -2 corresponds to y_0 = dab left and being normally-distributed around 2 corresponds to y_0 = dab right. On the contrary, sampling x requires the additional information the image encoding provides which helps resolve the relative nature of the learned spatial concepts - sample a joint angle trajectory which results in a left dab with respect to *what is present in i*.
>
> **Re. DMP baseline:** While a dynamic movement primitive seems like a good potential baseline, we are not aware of DMPs being compatible with multiple different demonstrations of the same task---ProMPs might be a good solution to that problem. However, mapping perturbations of DMP parameters to semantically-consistent perturbations (with respect to given user specifications) of robot joint trajectories is a non-trivial process, making any comparisons challenging.
>
> **Re. real robot experiments:**  To clarify, the output of the model are 7 DoF joint position (angles) and joint effort trajectories. The prediction performance of each model is not reported over a held-out set of trajectories but rather over a set of newly-sampled trajectories which are evaluated through a set of heuristics. Moreover, we demonstrate the suitability of the generated trajectories to be executed on the robot by using the robot kinematic model in order to translate the joint position part of each sample to corresponding end-effector movements and project those to the image plane of the robot’s camera sensor (see Appendices D and E). We choose to evaluate in this way to separate this paper’s contributions from issues of hybrid impedance control on the PR2 platform, and different ways of utilising both joint position and joint effort information within low-level control.
>
> We’ve additionally devised an experiment based around the task of pouring and different manners of executing it. This physical experiment is still ongoing, but the setup and preliminary description is in Section 7 and Appendix E of the revised manuscript. Videos of provided demonstrations can be seen on the supplementary website. If accepted, the camera ready version will include full results from this experiment.
>
> The experiment still follows the problem description in Section 2 -- namely some factors of variation describe where to pour (red or blue cups present in the image) and others how to pour (e.g. behind the target cup or on its side). The preliminary results analyse the usefulness of weak supervision through discrete labels for incorporating information from all input modalities. Complete analysis across all different baselines and demonstration concepts, as in section 6, will be in the final version of the paper.

---

### Official Review · AnonReviewer4 · 2020-10-29
**Experiments need to better establish the claims**

**Rating:** 7
**Confidence:** 4

**Review:**

*Summary*
Under the context of learning from demonstrations, the paper studies the problem of leaning interpretable low dimensional representations from high dimensional multimodal inputs using weak supervision. Paper argues that since robots and humans have different levels of abstractions and mechanisms, observation+action spaces between them are greatly misaligned which complicates learning by directly observing humans. However, the underlying concepts essential for tasks lie in a much lower-dimensional manifold. Learning this manifold effectively and in an interpretable way, especially using weak supervision, can significantly change how robots can acquire skills from demonstrations and generalize them to new unseen scenarios. Towards this end, the paper proposes to learn probabilistic generative models capturing high-level notions from demonstrations using variational inference. The strength of the paper is in demonstrating that conditional latent variable models can learn disentangled low dimensional represented using weak supervision; which authors effectively demonstrated using real-world experiments. My main reservations are in terms of the technical novelty of the paper and the narrow scope of experimental evaluation.

*Suggestions*
1. Use of variational inference for organizing high dimensional multi-modal inputs to low dimensional useful representations is already a well-established idea. However, the idea of learning "interpretable representations" using "weak supervision" is interesting and probably the most significant bit in this work. This however warrants appropriate experimental section probing styles/ strengths/ quality/ modality of supervision used. The current experimental section starts strong on the former (generating behaviors using concepts) but overlooks the latter, arguably the more interesting questions.
2. Authors motivate the paper using the idea that learning interpretable low dimensional concepts called "common sense" will enable generalization and adaptation. Experimental section fails to investigate and establish these claims.
3. Lastly, the experimental setup and the problem formulation around dabbing is very cleverly defined to communicate the ideas. This however comes across as too narrow. Additionally, the label groups for this problem cleanly separates into interpretable disjoint concepts. It's unclear if such interpretability and clear separation (hence the possibility of weak supervision) is true in general for common real-world problems. Paper makes no attempt to establish the generality of the approach.

---

> ### Author Response · Authors · 2020-11-24
> **Response to reviewer 4**
>
> We thank the reviewer for their comments and feedback.
>
> **Re. different styles of supervision:** We agree that studying the effect of different styles, quality and amounts of supervision on the eventual model performance are interesting questions. A potential avenue for future work is analysing how much correlation can be present in the captured data without hurting the generalisation capabilities of the model when composing the disentangled factors of variation. I.e. if the demonstrations contain only 3 out of 4 possible underlying behaviour combinations - e.g. “soft slow dab”, “soft quick dab” and “hard slow dab” - can the model still generate the 4th, never-seen one? What about having demonstrated only 2 out of 4 possible behaviour combinations. However, we’ve found that doing full justice to these questions requires a separate, in-depth study which is the focus of our current and future work.
>
> **Re. claims on generalisation:** Adaptation and generalisation are meant to refer to fact that perturbations in the high-level concept space (equivalent to changing the task specification - e.g. where to dab next) can be easily mapped to the corresponding low-level robot behaviour. The claims and motivation in the introduction will be revised to better-reflect the findings of the performed experiments.
>
> **Re. narrow experimental evaluation:** We’ve devised an additional experiment based around the task of pouring and different manners of executing it. This physical experiment is still ongoing, but the setup and preliminary description is in Section 7 and Appendix E of the revised manuscript. Videos of provided demonstrations can be seen on the supplementary website. If accepted, the camera ready version will include full results from this experiment.
>
> The experiment still follows the problem description in Section 2 -- namely some factors of variation describe where to pour (red or blue cups present in the image) and others how to pour (e.g. behind the target cup or on its side). The preliminary results analyse the usefulness of weak supervision through discrete labels for incorporating information from all input modalities. Complete analysis across all different baselines and demonstration concepts, as in section 6, will be in the final version of the paper.

---

### Official Review · AnonReviewer2 · 2020-10-31
**Novel idea**

**Rating:** 7
**Confidence:** 3

**Review:**

This paper presents a way to learn from demonstrations with weak or no labels. The premise behind this paper is that even when humans provide labels during a demonstration, those labels often do not fully describe the data (e.g., the human may say "soft" when "fast" would also apply). This paper presents a technique that uses latent variables to model the uncertainty over a group of class labels that could describe the task (e.g., slow, soft, left-of-object). The variables are modeled such that the observation is conditionally independent of the human provided labels given the latent variables. This allows the human provided labels to be decoupled (or disentangled as the paper calls it) from the observations. By doing so, it is possible to have only partial labels (weak labels). This model was applied to a task where a human would teleoperate a robot arm and apply a dabbing motion in relation to an object in the scene. The operator would provide only one of several possible applicable labels for each demonstration. The results show that the models using the weak labeling out-performed models with no labeling.

This paper proposes and interesting and novel way to handle weak labels from human demonstrators. By separating them, not only can they handle weak labels, but also multiple non-conflicting labels, or even no labels. This is an interesting contribution. The paper does a good job of comparing the methodology with 3 different off-the-shelf models for implementing the inference networks (GS, AAE, and VAE), as well as comparing each with and without the weak labels (baseline). There are several instances in the results where the weak label models far outperform the baseline models without weak labels (e.g., Table 3 VAE for "soft").

However, there are many instances where the baseline models outperform the weak-label models, although not as dramatically. Nonetheless, the results in Tables 2 and 3 seem inconsistent. While the weak-label models are usually better, there are several cases where the baseline outperforms. Furthermore the extremeness of some of the results are a bit concerning. It would be nice to see some further exploration of this. In particular, the choice of task and label groups could be significantly correlated with the performance of the models. Further experiments should be done with other, more standard tasks and potentially user-provided labels to better determine the performance of the weak-label models as compared to the baselines.

Overall, though, this paper does present a novel solution to the problem of user-provided labels. They're often not fully descriptive, which can leave some models confused (e.g., a demonstration can be "soft" but be labelled "fast"). The proposed solution attempts to solve this while also being probabilistically complete. While this certainly necessitates further experimental evidence to prove the usefulness of the model, the idea itself and preliminary evidence provided here are sufficient for publication.

---

> ### Author Response · Authors · 2020-11-24
> **Response to reviewer 2**
>
> We thank the reviewer for their comments and feedback.
>
> **Re. comments on results in Tables 2 and 3:** The latent spaces of all trained models can be used to condition on a single user label. It is only the semantically-aligned ones that can be used to compose multiple labels—see Eqs. (7) and (8) and Alg. 1 and 2 in Appendix B. The small difference in numerical results shows that all models have the capacity to represent the full set of demonstrated phenomena. However, unless otherwise optimised, they will be unintuitively entangled and not interpretable. So, even where quantitative results seem close, we show the benefit of using weak labels in Figure 5—we can only perform label composition effectively with a disentangled latent space.
>
> **Re. comments on additional experiments:** We’ve devised an additional experiment based around the task of pouring and different manners of executing it. This physical experiment is still ongoing, but the setup and preliminary description is in Section 7 and Appendix E of the revised manuscript. Videos of provided demonstrations can be seen on the supplementary website. If accepted, the camera ready version will include full results from this experiment.
>
> The experiment still follows the problem description in Section 2 -- namely some factors of variation describe where to pour (red or blue cups present in the image) and others how to pour (e.g. behind the target cup or on its side). The preliminary results analyse the usefulness of weak supervision through discrete labels for incorporating information from all input modalities. Complete analysis across all different baselines and demonstration concepts, as in section 6, will be in the final version of the paper.

---

### Decision · Program_Chairs · 2021-01-07
**Final Decision**

**Decision:**

Accept (Poster)

**Comment:**

The paper considers the problem of learning interpretable, low-dimensional representations from high-dimensional multimodal input via weak supervision in a learning from demonstration (LfD) context. To mitigate the disparity between the abstractions that humans reason over and the robot's low-level action and observation spaces, the paper argues for learning a low-dimensional embedding that captures the underlying concepts. The primary contribution of the paper is the ability to learn disentangled low-dimensional representations that are interpretable from weak supervision using conditional latent variable models.

The paper was reviewed by three knowledgeable referees, who read the author response and discussed the paper. The paper considers a challenging problem in learning from demonstration, namely dealing with the disparity that exists between the ways in which humans and robots model and observe the world, a problem that is exacerbated when reasoning over high-dimensional multimodal observations. As the reviewers note, the use of variational inference to learn low-dimensional interpretable representations from weak supervision is compelling. The primary concerns are that the contributions need to be more clearly scoped and that the experimental evaluation is a bit narrow. The authors make an effort to resolve some of these issues, in part through the inclusion of an additional experiment that considers pouring tasks. However, the extent to which this second task mitigates concerns about the narrow evaluation is not fully clear. The paper would be strengthened by the inclusion of experiments in a less contrived setting (and one for which the concepts are not necessarily disjoint) as well as a clearer discussion of the primary contributions.